# New Generation Pulse Modulation in Holmium:YAG Lasers: A Systematic Review of the Literature and Meta-Analysis

**DOI:** 10.3390/jcm11113208

**Published:** 2022-06-04

**Authors:** Antoni Sánchez-Puy, Alejandra Bravo-Balado, Pietro Diana, Michael Baboudjian, Alberto Piana, Irene Girón, Andrés K. Kanashiro, Oriol Angerri, Pablo Contreras, Brian H. Eisner, Josep Balañà, Francisco M. Sánchez-Martín, Félix Millán, Joan Palou, Esteban Emiliani

**Affiliations:** 1Department of Urology, Fundació Puigvert IUNA, 08017 Barcelona, Spain; alb.piana@gmail.com (A.P.); igiron@fundacio-puigvert.es (I.G.); akanashiro@fundacio-puigvert.es (A.K.K.); oangerri@fundacio-puigvert.es (O.A.); jbalana@fundacio-puigvert.es (J.B.); fsanchez@fundacio-puigvert.es (F.M.S.-M.); fmillan@fundacio-puigvert.es (F.M.); jpalou@fundacio-puigvert.es (J.P.); 2Department of Surgery, Universistat Autònoma de Barcelona, 08193 Barcelona, Spain; 3Department of Urology and Kidney Transplantation, Aix-Marseille University, APHM, Conception Academic Hospital, 13005 Marseille, France; michael.baboudjian@outlook.fr; 4Department of Urology, Hospital Alemán de Buenos Aires, Buenos Aires C1118 AAT, Argentina; pablocontreras.ar@gmail.com; 5Massachusetts General Hospital, Harvard Medical School, Boston, MA 02114, USA; beisner@mgh.harvard.edu

**Keywords:** holmium, laser, urinary stones, pulse modulation

## Abstract

(1) Background: New pulse modulation (PM) technologies in Holmium:YAG lasers are available for urinary stone treatment, but little is known about them. We aim to systematically evaluate the published evidence in terms of their lithotripsy performance. (2) Methods: A systematic electronic search was performed (MEDLINE, Scopus, and Cochrane databases). We included all relevant publications, including randomized controlled trials, non-randomized comparative and non-comparative studies, and in-vitro studies investigating Holmium:YAG lithotripsy performance employing any new PM. (3) Results: Initial search yielded 203 studies; 24 studies were included after selection: 15 in-vitro, 9 in-vivo. 10 In-vitro compared Moses with regular PM, 1 compared Quanta’s, 1 Dornier MedTech’s, 2 Moses with super Thulium Fiber Laser, and 1 compared Moses with Quanta PMs. Six out of seven comparative studies found a statistically significant difference in favor of new-generation PM technologies in terms of operative time and five out of six in fragmentation time; two studies evaluated retropulsion, both in favor of new-generation PM. There were no statistically significant differences regarding stone-free rate, lasing and operative time, and complications between Moses and regular PM when data were meta-analyzed. (4) Conclusions: Moses PM seems to have better lithotripsy performance than regular modes in in-vitro studies, but there are still some doubts about its in-vivo results. Little is known about the other PMs. Although some results favor Quanta PMs, further studies are needed.

## 1. Introduction

Urinary stone disease is a frequent condition associated with several comorbidities, such as diabetes and obesity, as well as environmental risk factors that are increasing in Western countries, with an estimated prevalence of 1–20% [1,2].

Technological advances led to the development of multiple therapeutic options for the treatment of urinary stones. The use of Holmium:Yatrrium-Aluminuim-Garnet (Ho:YAG) laser has gained a dominant role in endoscopic lithotripsy since its introduction in 1980 [3,4], becoming the most popular tool in this scenario [5,6,7]. Its popularity is justified by its ability to fragment stones of all compositions [8,9], the feasibility of employment with flexible devices [10,11], a good safety profile, and its versatility for soft tissue ablation [12,13].

The mechanics of the laser consist of the stimulation of holmium particles (inoculated on a YAG crystal) by a xenon or krypton light. This stimulation releases photons (light) of the same wavelength (2120 nm). The reaction is amplified in a mirrored optic box, and photons are emitted through a box opening in a pulsed fashion, which travel through an optic fiber. Once they reach the tip of the fiber and contact the liquid medium, a vapor bubble is generated through which the radiation travels and impacts the stone. Fragmentation occurs as a photothermal ablative mechanism with chemical decomposition [14]. 

Initially, the only laser settings that could be modified were the pulse energy and the pulse frequency, which were limited due to the use of low power lasers (<20 Watt). Subsequently, new technological advances allowed for changing the way in which the energy of each pulse was delivered, creating the concept of pulse modulation (PM). Initially, PM could be set in a short or long pulse (fast or slow energy delivery). 

More recently, in 2017, Lumenis^®^ (Yokne’am Illit, Israel) introduced a novel PM called Moses Technology, which consisted of modifying the shape of a pulse into two sub-pulses with different peak power. The first generates a vapor bubble through which the second sub-pulse travels and reaches the target, so that its energy does not dissipate in the medium [15]. The manufacturers provide two types of Moses PM: Moses Contact (MC) and Moses Distance (MD), the first to be used at 1 mm distance and the second at 2 mm distance [16].

Since 2017, other manufacturers have developed new pulse modulation techniques that are now available in the market. Examples of these are: Vapor Tunnel^TM^, Virtual Basket^TM^, Bubble Blast^TM^ (Quanta System, Samarate, Italy), Advanced Mode^TM^ (Dornier MedTech, Munich, Germany) and Stabilization Mode^TM^ (Olympus, Tokyo, Japan) [17]. In 2020, Lumenis released the Moses 2.0 system, which introduced a new PM called Optimized Moses. Instead of MC or MD, when using extended frequency (from 80 Hz up to 120 Hz) lithotripsy in high power lasers (≥120 W), Optimized Moses is the predefined PM setting [18,19]. 

Currently, urologists have wide access to these technologies, and it is their responsibility to choose which settings they use to perform more efficient and safe procedures. As different PMs will change bubble formation and energy delivery to the stone, it is plausible that they have a differential impact in lithotripsy performance. Manufacturers claim that the different technologies give better ablation stone rates, lower retropulsion, and smaller fragments, but evidence is not always present to support these claims, especially regarding newer PM technologies. In order to clarify this matter, we present a systematic review of the literature and meta-analysis focused on the new generation pulse modulation in Ho: YAG lasers for the treatment of urinary stones.

## 2. Objectives

Our objective is to conduct a systematic review of the literature and meta-analysis of available studies describing in-vitro and in-vivo new generation pulse modulation of Ho:YAG laser to determine their differences in terms of their lithotripsy performance, which includes: fragmentation efficacy, ablation ability, retropulsion, and fiber tip degradation for in-vitro studies; for in-vivo studies, stone-free rate, fragmentation, and operative time were assessed. 

## 3. Evidence Acquisition

### 3.1. Protocol 

This systematic review was conducted according to the principles highlighted by the European Association of Urology (EAU) Guidelines Office and the updated Preferred Reporting Items for Systematic Reviews and Meta-analyses (PRISMA) recommendations.

### 3.2. Search Strategy 

A literature search of studies published in English with no time restriction was conducted in March 2022 using PubMed/Medline, Scopus, and Cochrane databases. The literature search used both free text and MeSH terms. A manual search of bibliographies of included studies and previous systematic reviews was also performed. The search strategy used the combination of the following terms grouped according to the Boolean operators (AND, OR, NOT): “Holmium-YAG Laser”, “Laser”, “Holmium”, “Holmium-YAG”, “Pulse”, “Modulation”, “Moses”, “Moses 2.0”, “Optimized Moses”, “virtual basket”, “bubble blast”, “vapor tunnel”, “advance mode”, “stabilization mode”. 

### 3.3. Eligibility Criteria

A specific population (P), intervention (I), comparator (C), outcome (O), and study design (S) framework defined the study eligibility. Studies were considered eligible if they fulfilled the following criteria:(P): All in-vitro and in-vivo studies investigating the use of Holmium-laser lithotripsy performance employing any new pulse modulation settings: Moses and Moses 2.0 for Lumenis^®^ 120 W, Vapor Tunnel, Bubble Blast, and Virtual Basket (Quanta System, Samarate, Italy); Advanced Mode (Dornier MedTech, Munich, Germany) and Stabilization Mode (Olympus, Tokyo, Japan);(I): Performance of different pulse modulation settings in Ho:YAG laser lithotripsy;(C): Comparative and non-comparative studies;(O): In-vivo outcomes: stone-free rate, fragmentation. and operative time; in-vitro outcomes: ablation ability, fragmentation efficacy, retropulsion, and laser tip degradation;(S) Both prospective and retrospective studies were included.

### 3.4. Study Selection

Mendeley reference software removed duplicate records identified. Initial screening was performed independently by two investigators (A.S. and A.B.) based on the titles and abstracts of the article to identify ineligible reports. In case of duplicate publications, either the higher-quality or the most recent publication was selected. Reviews, meta-analyses, commentaries, abstracts of non-published studies, authors’ replies, theses, and case reports were excluded. Potentially relevant studies were subjected to a full-text review, and the relevance of the reports was confirmed after the data extraction process. Disagreements were resolved by consultation with a third co-author (P.D.). Figure 1 shows the flowchart depicting the overall review process according to the PRISMA statement recommendations.

### 3.5. Data Extraction

Data from the studies included in the review were extracted by two authors (A.S. and A.B.) in an a-priori developed data extraction form. The reliability and completeness of data extraction was crosschecked by another member of the review team (P.D.). We independently extracted the following variables from the included studies: -For in-vitro studies: first author’s name, publication year, type of laser and fiber size, pulse modulation, energy and frequency settings, fiber-stone distance, stone composition and hardness, experimental conditions, compared variables, and summarized results;-For in-vivo studies: first author’s name, publication year, study design, intervention, type of laser and fiber size, pulse modulation, energy and frequency settings, population, median stone size, stone Hounsfield units, operative time, fragmentation time, retropulsion, SFR, and complications.

When more than one article was based on the same study population, we included the most recent report. All discrepancies regarding data extraction between the three authors were resolved by consensus with a senior author (E.E.).

### 3.6. Assessment of Risk of Bias of Included Studies

The quality of the included studies and their design were considered according to the Jadad scale [20] for Reporting Randomized Controlled; we also used the Methodological Index for Non-randomized Studies (MINORS) [21] to assess the methodological quality of comparative and non-comparative studies. The former is a 5-point scale containing two questions for randomization, two for blinding, and one for the evaluation of dropouts. The latter is a 24-point and 16-point scale for comparative and non-comparative studies respectively, containing eight items for all non-randomized studies and four additional items for comparative studies, with a maximum score for each item of 2 and a minimal score of 0.

There are no validated instruments to assess the methodological quality of in-vitro studies, so we did not assess the quality of these studies.

### 3.7. Meta-Analyses for In-Vivo Studies

A meta-analysis was performed when two or more studies reported the same outcome under the same definition. For the computational part of the meta-analysis, various approaches were used to pool effect measures between studies. For continuous results, the mean and standard deviation (SD) were used. For post-operative complication and stone-free rate, we reported data as dichotomous events and calculated pooled odds ratios (ORs) and corresponding 95% confidence intervals (CIs). We used either a fixed- or a random-effect model for calculations of ORs according to the heterogeneity of the pooled studies. We assessed heterogeneity using the Cochrane Q-test and quantified it using I^2^ values. In case of heterogeneity (Cochrane Q-test *p* < 0.05 and I^2^ > 50%), we used a random-effect model, otherwise the fixed-effect model was used. All statistical analyses were performed using Cochrane Collaboration Review Manager software (RevMan v.5.4; Cochrane Collaboration, Oxford, UK). Statistical significance was set at *p* < 0.05.

## 4. Results

### 4.1. Literature Search and Inclusion Studies

A systematic review was conducted using the aforementioned databases; three authors participated in the literature search and data acquisition process (A.S., A.B., and P.D.). 

The initial search yielded 203 records (Figure 1). After the identification of these studies through database searching, 167 records were excluded. A total of 37 studies were screened and fully reviewed, excluding 13 studies that did not comply with the selected outcomes or had insufficient data, with a final selection of 24 studies: 15 in-vitro and 9 in-vivo. Table 1 and Table 2 summarize the in-vitro and in-vivo studies characteristics, respectively, including outcome measures.

### 4.2. Description of the Studies 

#### 4.2.1. In-Vitro Studies

Table 1 shows a detailed description of the results of the in-vitro studies, listing the laser type, the pulse modulation, the energy and frequency settings, the fibers and stones used, the experimental conditions, the compared variables, and the main results. A total of 15 articles were included. 

Three main types of experiments were performed: (1) static laser activation, (2) automatized movement of the laser fiber tip across the stone surface to generate a cut or ablation surface, and (3) lithotripsy experiments performed by urologists. 

King [22] and Ballesta et al. [32] used static laser activation for crater formation. Black et al. [26] also used a static laser activation, but the fiber and the stone were placed inside a spherical test tube in order to mimic the popcorn dusting during retrograde intrarenal surgery (RIRS). For automatized movement of the laser fiber tip, Aldoukhi et al. [23] performed 2-cm linear cuts on the stone surface. To mimic a “painting” dusting lithotripsy technique, Winship et al. [24] displayed an automatized moving laser-holding arm that, with a spiral motion, generated a square lithotripsy area in the stone surface. Finally, lithotripsy experiments performed by urologists were set in different scenarios. Ibrahim, Keller, and Khajeh et al. [9,27,28] used artificial recipients to model the urinary tract, whereas Jiang et al. [31] performed retrograde intra-renal surgery (RIRS) in extracted porcine kidneys after stone placement in the renal pelvis with pyelothomy. Elhilali et al. [16] performed RIRS in living pigs under general anesthesia. 

The vast majority of the articles analyzed differences between Moses PMs (MC and/or MD) and regular Lumenis short and long pulse modulation. Only three articles compared Moses PM with other lasers [17,31,32]. 

Jiang et al. [31] compared Moses PM in a Lumenis Pulse™ 30 H Ho:YAG laser with Lumenis Ho and Nd:YAG VersaPulse PowerSuite™ laser regular modes and super-thulium fiber laser (sTFL) (IPG Photonics). They observed that sTFL generated smaller fragment remnants than after conventional and Moses Ho:YAG RIRS in a porcine kidney model. Terry et al. [17] compared Lumenis regular and Moses PM and Quanta’s Vappor Tunnel (VT), SP short and long pulses. In their experiments, Lumenis MD had better lithotripsy performance in terms of crater and cut volume, whereas MC was clearly inferior. Ballesta et al. [32] tested Quanta’s VB, VT, and Bubble Blast (BB) PMs observing that, at high power settings (2 J × 30 Hz), VB had the greatest lithotripsy ablation rate, whereas VT had the lowest rate in hard stones. 

When comparing different PMs, a wide range of variables were analyzed, which varied especially according to the type of experiment that was performed. The most frequently reported variables were those regarding ablated volume or mass. For Moses technology, results favored MC and MD against SP (King et al. [22]) and against both regular modes (SP and LP) (Elhilali et al. [18]). Aldouki et al. [23] found that MD had the highest ablation and fragmentation rates both at 0 and 1 mm distance when compared to the rest of the Lumenis’ PMs. In another article, the same authors found better stone ablation with MD than with SP, especially at a fiber moving speed of 3 mm/s rather than 1 mm/s [25]. 

Terry et al. [17] found a better lithotripsy performance of MD compared to the rest of Lumenis’ and Quanta’s PMs. They also found MC to be the worst in this category. Winship et al. [24] also found MD to have the greatest ablation performance in soft stone at 1 mm distance but was as good as the regular modes (SP and LP) when in contact. In contrast, MC had the greatest ablation of soft stones. No difference in ablation was found between PMs on hard stones at any distance, nor for soft stones for distances > 1 mm.

In the study by Ventimiglia et al. [30], stone ablation was equal between Moses and long pulse, and both were inferior compared to sTFL. When analyzing Quanta’s PMs, Ballesta et al. [32] found Virtual Basket at 2 J and 30 Hz was the combination with the greatest ablation rate compared to the three available PMs, whereas VT had the lowest ablation rate in hard stones. Finally, when Dornier’s PMs were studied, Ho et al. [33] found higher crater volumes with shorter PMs. 

When assessing RIRS in-vitro, Elhilali et al. [16] found no procedural nor lasing time difference when using Moses PMs when compared to regular mode, whereas Ibrahim et al. [27] reported less procedural time, a reduced number of pedal laser activation, and a higher percentage of lasing time vs. pausing time with MC. 

Regarding residual fragments, Black et al. [26] reported smaller fragment size distribution with MD for both 20 W and 40 W settings, except for high frequency (80 Hz) in a popcorn RIRS model. When comparing Lumenis regular modes to Moses, Khajeh et al. [28] found no difference between the percentage of smallest fragments (<0.25 mm), but MC and MD produced a greater mass of fragments <2 mm compared to LP.

#### 4.2.2. In-Vivo Studies

Nine studies met inclusion criteria, out of which seven were comparative, including one RCT, one prospective study, and the rest were retrospective studies. The remaining two studies were noncomparative observational series. A total of 1482 patients participated in the included studies.

Regarding the new generation pulse modulation used, eight studies used Moses technology (Lumenis), and one study used Virtual Basket (Quanta). In most of the studies, the surgery performed was ureteroscopy (*n* = 5); two studies included both ureteroscopy and RIRS. The intervention performed in the two observational noncomparative studies were mini- and ultra-mini-percutaneous nephrolithotomy (PCNL). Stone dimensions and Hounsfield units among the included studies varied widely.


**Comparative Studies**


Seven out of eight comparative studies included operative time as one of the main outcomes; there was a statistically significant difference in favor of new generation pulse modulation technologies in six of these studies. The study by Knoedler et al. [35] published in 2022 found no statistically significant differences; the authors state that bias could have been introduced due to the possibility of changing of laser settings across cases, relying on the accuracy of electronic medical records. In addition, the clinicians were not blinded, suggesting that the advantages of Moses mode turned out to be less noticeable.


*Fragmentation time*


This outcome was assessed in six studies; in five of them, a statistically significant difference was found in favor of the new generation PM technologies (i.e., Moses and Quanta system). The study by Knoedler et al. [35] did not find a statistical difference in terms of fragmentation time. 


*Retropulsion*


Only two studies reported retropulsion as a secondary outcome. Ibrahim et al. [18] in 2020 published a prospective double-blinded RCT comparing regular and Moses modes of Ho:YAG lithotripsy; the authors measured retropulsion using a Likert scale from 0 (no retropulsion) to 3 (maximum retropulsion) and found that the Moses technology had a mean grade of 0.5 vs. 1 (*p* = 0.01) with the regular Holmium laser. Bozzini et al. [34] published in 2021 a prospective comparative study using the Virtual Basket technology with the regular mode and reported less retropulsion using Virtual Basket. 


*Stone-free rate (SFR)*


All included studies measured SFR, although the definitions were different among these studies. Seven studies assessed SFR at 1 month’s follow-up; in the rest it was evaluated at 2–4 months. There was a trend towards no statistically significant differences between the new PM technologies and the regular laser modes.


*Complications*


In terms of complications classified using the Clavien–Dindo scale, there was no statistically significant difference between the new PM technologies and the regular laser modes.


**Non-Comparative Studies**


Two observational non-comparative studies were included. Reddy et al. [39] in 2021 published a prospective case series of 110 patients who underwent mini PCNL using Moses p120 W; they found that the use of Moses technology in conjunction with a proper suction system may potentially achieve maximum dusting, improving SFR. In a similar way, Leotsakos et al. [40] published their series of 12 patients who underwent ultra-mini PCNL using the same technology. 

### 4.3. Risk of Bias of Included Studies

The results of the methodological quality assessment are included in Table 2. There was only a double-blind RCT by Ibrahim et al. [18] in 2020; using the Jadad scale, a score of 5 out of 5 points was given. The mean score for non-randomized comparative studies (*n* = 6) was 15.5 (range 12–22 out of 24); the highest score was given to the study by Bozzini et al. [34] in 2021. For non-comparative studies (*n* = 2), the mean score was 10.5 (range 9–12 out of 16).

### 4.4. Meta-Analyses

Four studies provided data on the association between SFR and PM. The forest plot (Figure 2) revealed no significant difference between the Moses and the regular pulse (OR 1.15, 95% CI, 0.80–1.65, *p* = 0.46). The Cochrane’s Q and I^2^ tests revealed no significant heterogeneity. 

Three studies provided data on the association between operative time and pulse modulation. The forest plot (Figure 3) revealed no significant difference between the Moses and the regular pulse (mean difference −15.76 min, 95% CI, −45.97 to 14.44; *p* = 0.31). The Cochrane’s Q (*p* < 0.001) and I^2^ (98%) tests revealed significant heterogeneity. 

Two studies provided data on the association between fragmentation time and pulse modulation. The forest plot (Figure 4) revealed no significant difference between the Moses and the regular pulse (mean difference −1.71 min, 95% CI, −11.81 to 8.38; *p* = 0.74). The Cochrane’s Q (*p* = 0.02) and I^2^ (81%) tests revealed significant heterogeneity.

Four studies provided data on the association between post-operative complication and pulse modulation. The forest plot (Figure 5) revealed no significant difference between the Moses and the regular pulse (OR 0.63, 95% CI, 0.34–1.15, *p* = 0.13). The Cochrane’s Q and I^2^ tests revealed no significant heterogeneity. 

## 5. Discussion

In 2017, Lumenis Moses Technology inaugurated the new pulse modulation era, followed by the other pulse modes developed by Quanta System, Dornier MedTech, and Olympus. As our systematic review shows, most of the studies, both in-vitro and in-vivo, have analyzed Moses PM. This is explained by the fact that it was the first new-generation PM, and many authors wanted to test how this possible game-changing tool could and should be applied.

Surprisingly, no articles studying the Olympus’ Stabilization Mode were found to support the commercial claim that this PM produces a vapor tunnel bubble that clears a path through the water to reduce retropulsion [41].

Regarding Advance Mode (Dornier MedTech), only one in-vitro article was found [33]. Marketing material distributed by the company on the website claims that AM reduces stone movement during lithotripsy and patient treatment time [42]. However, Ho et al. [33] showed that Fragmentation Mode (with a total width at half of the maximum pulse duration of 75 μs) generated larger craters than the Advance Mode (with a longer total width at half of the maximum pulse duration of 200 μs). It seems that the only difference between these pulse settings is pulse length, and no information is available regarding their pulse shapes or associated retropulsion.

As noted above, Lumenis Moses PM is the most represented in the reviewed studies, especially when comparing Moses to regular short and long pulses. Before Moses technology, studies had shown that regular short pulse length was more ablative than long pulse [43,44,45]. Despite this apparent lower efficiency, long pulse has shown to produce less fiber tip degradation and stone retropulsion in-vivo and in-vitro [46,47], as well as generation of smaller fragments [48], being a more suitable option for dusting stone strategies. However other studies did not show significant differences in fragmentation efficiency or fragment sizes [49]. 

Regarding in-vitro studies, several articles favored Moses PM over regular pulse modes in terms of ablated volume or mass (Elhilali [16], Ibrahim [27], Khajeh et al. [28]). Less retropulsion was also associated with Moses PMs when compared to regular PM in the two studies that compared them [16,27]. Regarding residual fragments, Black et al. [26] reported smaller fragment size distribution with MD (for 20 W and 40 W settings, except for high frequency at 80 Hz) in a popcorn RIRS model. In addition, Khajeh et al. [28] found no difference between the percentage of smallest fragments (<0.25 mm), but MC and MD produced a greater mass of fragments < 2 mm compared to LP. Fiber tip degradation was lower with Moses PM in Khajeh’s study [28] compared to regular PM, whereas no differences were observed in Winship’s [24] study. 

As for in-vivo studies, the majority compared Lumenis Pulse™ 120 H to regular modes. The comparison of Moses technology versus regular modes was also assessed by Corsini et al. [50] in a narrative review published in 2022. This study affirms that the Moses effect, although widely characterized, lacks strong clinical evidence of its superiority in surgical outcomes, in particular compared to Long Pulse, as they have similar peak power, total power width, and comparable ablation efficiency and retropulsion as stated by Winship et al. [24]. Similarly, in our meta-analysis performed using included in-vivo studies that compared the use of regular PM vs. Moses PM, no statistically significant differences were found in terms SFR, operative and lasing times, or complication rates. Thus, we agree with these authors that there is no clinical certainty that Moses PM substantially improves lithotripsy performance, and that the studies available to date have several limitations and biases. However, the highest methodological quality study published by Ibrahim et al. [18], a double-blind RCT, found that operative time, fragmentation time, and retropulsion were superior in the Moses mode, although SFR and complications did not show statistical differences.

Quanta PMs were the second most studied after Lumenis Moses PM, but with a low number of published studies. Ballesta et al. [32] compared VB, VT, and BB in-vitro. They found Virtual Basket to be more ablative than Bubble Blast in both 1 J × 60 Hz and 2 J × 30 Hz, with the latter combination achieving the highest ablation. They concluded that, in high-power settings, Virtual Basket accounted for highest ablation rates, exceeding Bubble Blast. On low-power settings, Virtual Basket was superior to Vapor Tunnel. These findings happened with both soft and hard stone phantoms. From our initial selection of articles, Vizziello et al. [51] presented at the 36th World Congress of Endourology an abstract of an in-vitro study of an endoscopic laser cystolithotripsy in a synthetic bladder model, comparing VB with Standard Mode, reporting less subjective retropulsion. 

Terry et al. [17] compared Quanta’s VT, SP, and LP with Lumenis MC, MD, SP, and LP in-vitro. The MD mode was superior in ablative measures than the rest of the parameters. Laser-stone distance had a negative impact on ablation in all PMs except for VT, which maintained a much greater proportion of its 0.5 mm ablation efficacy once SD increases to 2 mm. 

A prospective comparative study of VB versus the regular mode using Quanta System Ho:YAG also found a trend towards better performance of this new generation PM in similar outcomes. These findings lead us to believe that there may be some advantages to using the Moses and VB technologies, which may have an impact on our daily practice.

In this direction, in a published video (excluded from the systematic review), Bozzini et al. [52] presented a randomized controlled trial comparing endoscopic retrograde endoscopic surgery using Quanta VT versus regular dusting mode (105 patients per group). They reported less procedural time, less retropulsion, no needs of basket stone retrieval, and no ureteral lesions with the Vapor Tunnel mode.

High-power and high-frequency Ho:YAG laser from Lumenis is now able to use frequencies higher than 80 Hz and up to 120 Hz. This new technology is called Moses 2.0 and has a predefined pulse modulation setting called Optimized Moses when extended frequencies are being used. When utilizing >80 Hz, the option to select MD or MC modes is replaced with a system-provided optimized Moses mode using rapid pulse sequence [53].

Only Majdalany et al. [36] have investigated Moses 2.0 compared to regular Moses PM (or Moses 1.0). They retrospectively compared the use of Moses 2.0 versus 1.0 in patients that underwent f-URS for solitary stones performed by a single surgeon, showing a SFR of 71% and 90% for Moses 1.0 and 2.0, respectively. No additional information was found regarding Moses 2.0 shape and or bubble formation and interaction. Therefore, it is difficult to state if any changes in lithotripsy performance would be due to Optimized Moses PM rather than the use of higher frequency settings. The use of frequencies, up to 80 Hz for stone dusting, have shown to break stones into fine fragments [54], but little is known about the performance when extended frequencies (80–120 Hz) are used.

Thulium Fiber Laser (TFL) has recently been introduced as a endoscopic lithotripsy tool that directly competes with Ho:YAG laser. TFL has a greater water absorption peak, which corresponds to a low threshold for tissue ablation and stone lithotripsy compared to Ho:YAG laser, and it has the ability to function at very low energies and extremely high frequencies [55]. Multiple studies have reported a more efficient lithotripsy [56]. TFL has also show its advantages with respect to Ho:YAG laser in high evidence clinical studies both for shorter operative time [57] and higher stone free rate and less complications after flexible ureteroscopy [58]. In our review, two in-vitro studies pointed into the same direction [30,31]. 

An important issue that raises concerns are thermal lesions of the urinary system during laser lithotripsy that can lead to ureteral stricture [59]. Temperature rise because of laser activation can generate tissue damage, especially when surpassing a 43 °C threshold temperature that leads to protein coagulation and tissue injury [60]. Temperature rise depends on laser power, exposure time, and fluid irrigation. Some recommendations are to keep low power settings (<10–15 W), to use “high-frequency” settings with caution, to perform intermittent laser activity, and to ensure fluid irrigation during the procedure [61]. In this review, an in-vitro study performed by Winship et al. [29] was included, which studied temperature changes while the laser was activated inside a ureteral access sheath (UAS) with different Lumenis PMs. They found that LP laser settings produced the greatest heat at most tested energy/frequency combinations, whereas Moses PMs produced the least heat at settings of 10 W of power. Nonetheless, at ≤10 W, temperature did not surpass the threshold for injury. Thermal dose reached potentially dangerous levels at 1 J × 20 Hz, but without significant differences between pulse types. MC was the only PM that exceed the risk temperature threshold by a small margin at 0.2 J × 70 Hz, although this was not statistically significant.

### Strengths and Limitations

To date, this is the first systematic review and meta-analysis analyzing new generation PMs, including pulse modulation other than Lumenis Moses Technology. 

Generally, in-vitro studies aim to recreate some aspects of the in-vivo laser performance. More mass or ablated volume or less retropulsion could translate into faster or more efficient lithotripsy. Unfortunately, in-vitro results should be taken with caution because there are many factors influencing in-vivo lithotripsy that are not being considered in in-vitro experiments. Nevertheless, the most reproducible in-vitro studies would be those that recreate lithotripsies performed by clinicians in animal or artificial models.

Moreover, high heterogenicity was found when assessing in-vitro studies. Different experimental conditions, used stones, laser-stone distance, fiber sizes, and ways to calculate the reported parameters make it difficult to compare different studies. Some experimental models, such as static laser positioning or automatized laser movement, are far from the laser-stone interaction that happens during endoscopic surgery, and should be interpreted as trends and/or preclinical studies. Additionally, the use of stone phantoms with different materials adds more heterogenicity when comparing different materials and different plaster-water ratios. BegoStone (the most commonly used material) has been validated for its acoustical properties for shock wave lithotripsy research [62] but not with laser lithotripsy.

In this review none or very few articles assessing new Ho:YAG pulse modes such as Quanta’s, Dornier MedTech’s, and Olympus’ PMs were found. Therefore, further studies are needed to issue statements about their performances.

## 6. Conclusions

Moses PM is the most studied new PM for being the first to inaugurate this new era. Most in-vitro studies grant better lithotripsy performance to Moses PM in terms of shorter operative and fragmentation time and retropulsion when compared to regular PM. However, when we meta-analyzed in-vivo studies, no statistically significant differences were found in terms of SFR, operative and lasing times, or complication rates. Thus, it is still unclear if Moses is really superior to regular Long Pulse, and more high quality in-vivo studies are required.

Very little is known about other PMs, especially with Dornier’s Advance Mode, Olympus’s Optimized Mode, and Lumenis’ Moses 2.0. Some research has been done on Quanta PM, but its promising clinical results still have to be validated with further studies, especially comparing lasers and PM from different companies. Longer follow-up would help recognize potential long-term complications associated with endoscopic surgery such as ureteral strictures.

## Figures and Tables

**Figure 1 jcm-11-03208-f001:**
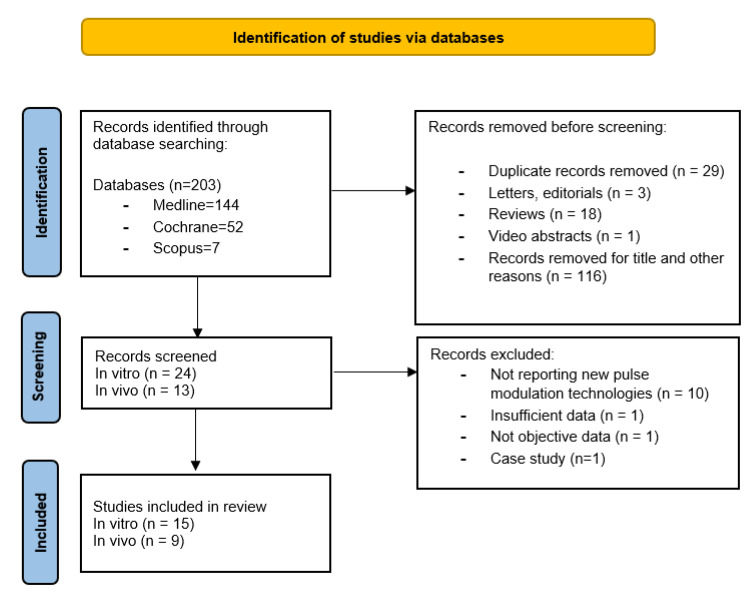
PRISMA (preferred reporting items of systematic reviews) flow diagram of study inclusion process.

**Figure 2 jcm-11-03208-f002:**
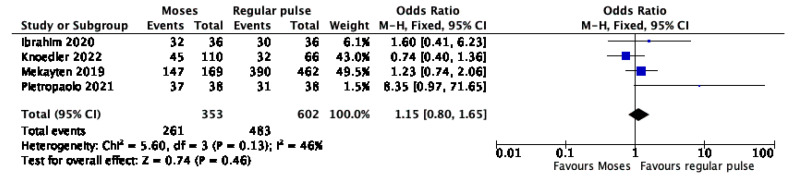
Forest plot of the meta-analysis comparing stone free rate between Moses versus regular pulse modulation [18,35,37,38].

**Figure 3 jcm-11-03208-f003:**
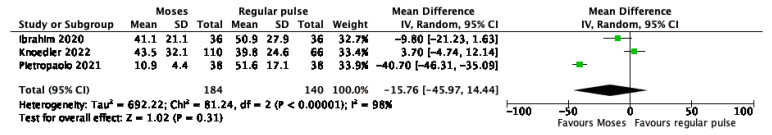
Forest plot of the meta-analysis comparing operative time between Moses versus regular pulse modulation [18,35,37].

**Figure 4 jcm-11-03208-f004:**
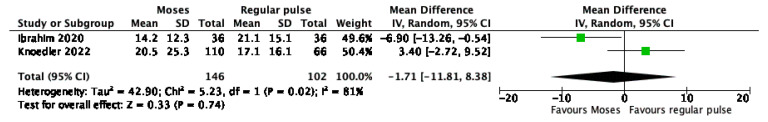
Forest plot of the meta-analysis comparing fragmentation time between Moses versus regular pulse modulation [18,35].

**Figure 5 jcm-11-03208-f005:**
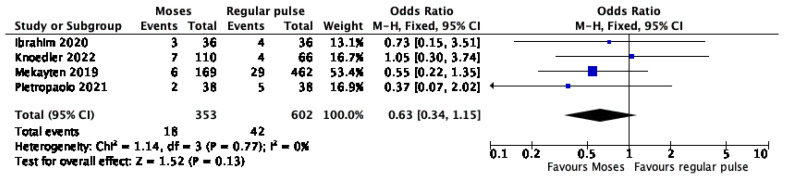
Forest plot of the meta-analysis comparing post-operative complications between Moses versus regular pulse modulation [18,35,37,38].

**Table 1 jcm-11-03208-t001:** Study details of the included in-vitro studies.

Author, Year	Laser Used	Pulse Modulation Setting	Laser Setting (Energy J, Frequency Hz)	Fiber Size (μm)	Fiber-stone Distance (mm)	Stone Composition	Hardness (Plaster: Water)	Experimental Conditions	Compared Variables	Summarized Results
King 2021 [22]	Lumenis Pulse™ 120 H (Lumenis)	SP vs. MD	−1 J single pulse −0.5 J two pulses separated by at least 2 s	200	1	BegoStone and human COM, MAP and UA	Hard BegoStone (15:3)	Static laser activation for crater formation in: (1) Dry stones in air (2) Wet stones in air (3) Wet stones in water	-Crater volume -Energy transmission through 1 mm water -Cavitation bubble collapse pressures	Larger craters with MC and MD than with SP. All PMs had high energy transmission through 1 mm of water. No-Moses PMs generated much higher peak pressures.
Aldoukhi 2019 [23]	Lumenis Pulse™ 120 H (Lumenis)	SP vs. LP vs. MC vs. MD	1 J, 10 Hz	230	0 0.5 1 2	BegoStone	Hard (15:3) for static crater experiment and soft (15:5) for moving laser cut	Static laser activation for crater formation and automatized moving laser-holding arm performing a 2 cm linear cut	-Ablation crater volume -Ablated mass (difference between phantom weight before and after laser action)	More LSD distance, less ablation. No ablation from 3 mm distance. Greatest ablation was achieved with MD at 1 mm. Greater fragmentation at both 0 and 1 mm compared to other PMs.
Winship 2018 [24]	Lumenis Pulse™ 120 H (Lumenis)	SP vs. LP vs. MC vs. MD	0.4, 70 Hz	365	0 1 2	BegoStone	Hard (15:3) and soft (15:6)	Automatized moving laser-holding arm performed square lithotripsy in a spiral motion	-Ablated mass -Fiber tip degradation	Less LSD, greater ablation regardless of stone composition and pulse modulations. No ablation difference between PMs on hard stones at any distance. In contact with soft stones, MC produced the greatest ablation. At 1 mm from soft stones, MD produced the greatest ablation. No differences in fiber tip degradation.
Aldoukhi 2020 [25]	Lumenis Pulse™ 120 H (Lumenis)	LP vs. MD	0.5 J, 20/40/80 Hz	230	0	BegoStone	Soft (15:5)	Automatized moving laser-holding arm cutting phantoms at 1 and 3 mm/s	-Crater depth -Crater area -Ablation volume	Ablation MD > LP, specially at 3 mm/s MD has deeper craters than LP
Elhilali 2017 [16]	Lumenis Pulse™ 120 H (Lumenis)	Regular Mode * vs. MC and MD	Fragmentation: 0.8 J, 10 Hz and 1.5 J, 10 Hz	200 365	1 for MC 2 for MD	Plaster of Paris (gypsum) stones and AU 3000	UA 3000 (4:1)	-Static laser horizontal firing for retropulsion analysis -Automatized moving laser-holding arm performing a linear cut -Human performed RIRS for pyelic stone anesthetized pigs	-Stone retropulsion -Ablation volume -Surgeon subjective retropulsion	Less retropulsion and higher ablation volume in MC and MD compared to regular mode. Biggest difference with low energy with high frequencies (stone dusting regimes) together with larger diameter fibers. Less subjective retropulsion with Moses PMs but no differences regarding lasing and procedural times.
			Dusting: 0.5 J, 50 Hz							
Black 2020 [26]	Lumenis Pulse™ 120 H (Lumenis)	SP vs. MD	1 J, 20 Hz 0.5 J, 40 Hz 1 J, 40 Hz 0.5 J, 80 Hz	230	2	BegoStone	Hard (15:3)	Static laser pop-corn RIRS model in spherical test tube (repositioning the fiber in the center each 15 s)	-Fragment size distribution -Fragment mass lost in fluid	At 1 J × 20 Hz, MD created smaller fragments than SP. MD created the smallest fragment size distribution for both 20 W and 40 W settings, except for the high frequency 80 Hz
Ibrahim 2018 [27]	Lumenis Pulse™ 120 H (Lumenis)	SP vs. MC	For fragmentation: 0.8 J, 10 Hz	200	0	UA 3000	Soft (4:1)	Human performed lower pole RIRS in an artificial urinary trat model	-Subjective stone retropulsion -Time to compete dusting -Lasing time -Number of pedal uses -% laser on vs. Off -Total energy required	MC: Less procedural time both in fragmentation and pulverization, less subjective retropulsion, reduced number of times the pedal was pressed and higher percentage of time lasing vs. pausing.
			For dusting 0.4 J, 50 Hz							
Keller 2018 [9]	Lumenis Pulse™ 120 H (Lumenis)	LP vs. MC	0.2 J, 40 Hz	200	-	Human COM, COD, UA, CA, MAP CA, MAP, BR and CYS	-	Human performed RIRS model inside a 10 mm diameter glass container	-Morphology of dust and residual fragments	MC: more pronounced disruption of morphological characteristics of COD, MAP and CYS. Areas with hexagonal plate-like surfaces appeared on residual fragments and dust from BR
Khajeh 2022 [28]	Lumenis Pulse™ 120 H (Lumenis)	SP vs. LP vs. MC vs. MD	0.5 J, 30 Hz	230	-	Canine COM stones	-	Human performed RIRS model in a 20 mm inner diameter spherical 3D printed calyceal model	-Residual fragment size distribution -Fiber tip degradation	No difference between fragments <0.25 mm rate. MC and MD produced a greater mass of fragments <2 mm compared to LP Less fiber tip degradation with MC and MD than with SP.
Winship 2019 [29]	Lumenis Pulse™ 120 H (Lumenis)	SP vs. LP vs. MC vs. MD	0.6 J, 6 Hz 0.8 J, 8 Hz 1 J, 10 Hz 1 J, 20 Hz 0.2 J, 70 Hz	365	-	No stone used	-	Static laser activation inside a UAS (URS model)	-Mean temperature change from a baseline adjusted to 37 °C at 1 s and every 5 s -Cumulative equivalent minutes at 43 °C	At 1/10 Hz no thermal injury threshold was reached. LP generated the greatest temperature increase, but not statistically significant. Only MC at 0.2 J/70 Hz exceed the threshold by a small margin although this was not statistically significant
Ventimiglia 2020 [30]	sTFL (Urolase SP)	Regular	0.2–2 J, 8–80 Hz ~16 W combinations	200	0	BegoStone	Hard (15:3)	Static laser crater formation	-Retropulsion -Crater volume -Pulse shape	Retropulsion: Lowest with sTFL, highest with SP. LP = Moses Stone ablation: sTFL > Ho:YAG. LP = Moses LP had longer pulse and lower peak power than SP SP mode had the shortest pulse width and highest peak power
		Dual phase								
	Lumenis Pulse™ 120 H (Lumenis)	SP LP MC		230						
Jiang 2021 [31]	sTFL (IPG Photonics)	-	0.2 J, 80 Hz	230	-	Human calcium oxalate stones	-	Human performed pyelic RIRS in a porcine kidney with and without UAS and with or without continuous aspiration	-Stone clearance rate (SCR) -Residual fragment size distribution	Highest SCR with sTFL with UAS and aspiration. Lowest SCR with Ho:YAG laser without UAS and no aspiration. sTFL resulted in smaller stone remnants compared to Ho:YAG and Ho:YAG-MOSES. All groups had similar proportion of stone remnants <100 microns. The use of UAS improved SCR regardless of the type of laser used or use of aspiration
	Ho and Nd:YAG VersaPulse PowerSuite™ (Lumenis)	Regular Mode *	0.4 J, 40 Hz	420						
	Lumenis Pulse™ 30 H (Lumenis)	Moses MP *	0.2 J, 80 Hz	408						
Terry 2021 [17]	Lumenis Pulse™ 120 H (Lumenis)	SP LP MC MD	0.4 and 1 J,	272	0.5 1 2	BegoStone	Soft (15:6)	-Static laser crater formation -Automatized laser-holding moving arm generating a linear cut	-Crater depth -Crater area -Crater volume -Cut volume -Pulse duration	Ablation volume was different in all PMs. Lumenis MD had better lithotripsy performance. Lumenis MC was clearly inferior. VT maintains a much greater proportion of its 0.5 mm ablation efficacy once SD increases to 2 mm.
	Litho 100 High Power (Quanta System)	SP LP VP								
Ballesta 2021 [32]	Cyber Ho 150 W (Quanta System)	VB	0.5 J, 20 Hz 1 J, 60 Hz 2 J, 30 Hz	365	-	BegoStone	Hard (15:3) and soft (15:6)	Static laser crater formation in saline media	-Ablation rate (difference between stone weight before and after lithotripsy /lithotripsy time -Laser activation time to reach 3 kJ	Greatest ablation rate combination: VB, 2 J, 30 Hz. Lowest ablation rate: VT in hard stones. Ablation rates for VT and BB improved with increasing laser power. For hard stones, VB and BB had better performance with 2 J × 30 Hz than 1 J × 60 Hz. In low-power lithotripsy (10 W= 0.5 J × 20 Hz) ablation rate was higher with VB than VT
		VT	0.5 J, 20 Hz							
		BB	1.2 J, 10 Hz 1.2 J, 50 Hz 2 J, 30 Hz							
Ho 2021 [33]	H Solvo 35 W (Dornier MedTech)	FM vs. SM vs. AM	0.8 J, 10 Hz	365	0.5 1 2	Human COM and BegoStone	Hard (hCOM) and soft (BegoStone 15:6)	Static laser crater formation in air and saline media	-Crater volume -Maximum crater depth -Crater area -Dynamic of crater formation and its relation with the bubble formation	Longer pulse durations (AM) result in greater laser energy delivery to the stone Shortest PM (FM and RM) had higher crater volumes predominantly by wider craters. Crater depths were comparable among PMs

Abbreviations: SP: short pulse, LP: long pulse, MC: Moses Contact, MD: Moses Distance, RIRS: retrograde intrarenal surgery, URS: ureteroscopy, LSD: laser-stone distance, UAS: ureteral access sheath, BR: brushite, CA: carbapatite, COD: calcium oxalate dihydrate, COM: calcium oxalate monohydrate, CYS: cystine, MAP: magnesium ammonium phosphate, UA: uric acid, FM: Fragmentation Mode (full width at half maximum 75 μs), SM: Standard Mode (FWHM 150 μs,) AM: Advanced Mode (FWHM 200 μs), PM: pulse modes/modulations, VT: Vapor Tunnel, VB: Virtual Basket, BB: Bubble Blast, sTFL: super Thulium Fibre Laser, SD: Stone distance. * Does not specify if short or long pulse.

**Table 2 jcm-11-03208-t002:** Study details of the included in-vivo studies.

Author, Year	Study Design	Intervention	Laser	Pulse Technology	Fiber (μm)	Laser Setting	Population, n	Median Dimension, mm (HU)	Operative Time, min	*p* Value	Fragmentation Time, min	*p* value	Retropulsion (Mean Grade LIKERT Scale 0–3)	*p* Value	Stone Free Rate Definition	Stone Free Rate, %	*p* Value	Complications	*p* Value	Jadad Scale	Minors Scale (0–24)
						(Energy J, Frequency Hz)															
**Comparative studies**																					
Ibrahim 2020 [18]	RCT	URS	Lumenis Pulse™ 120 H	Moses	200	Dusting: 0.4 J, 80 Hz	36	1.7 (991)	41.1	0.03	14.2	0.03	0.5	0.01	3 month	88.4	*p* > 0.05	8.3%	>0.05	5	
				Regular mode		Fragmentation: 1.0 J, 10 Hz	36	1.4 (841)	50.9		21.1		1.0			83.3		11.1%			
Bozzini 2021 [34]	Prospective Comparative	RIRS	Quanta System Cyber Ho 100 W	VirtualBasket	272	0.6–1.0 J, 15 Hz	40	15.5	52.4	<0.05	19.8	<0.05	0	-	1 month	92.5					22
				Regular mode			40	16.2	67.1		28.7		3			77.5					
		URS		VirtualBasket	365	0.6–1.4 J, 10 Hz	40	11	35.7	<0.05	16.1	<0.05	0			87.5					
				Regular mode			40	12	49		20.4		3			92.5					
Knoedler 2022 [35]	Retrospective Comparative	URS/RIRS	Lumenis Pulse™ 120 H	Moses	200	Dusting: 0.3 J, 80 Hz	110	11.8	49.7	0.195	20.5	0.305	-	-	1 month	52.3	0.143	6.4%	0.936		13
				Regular mode		Fragmentation: 0.8 J, 8 Hz	66	11.6	39		17.1					65.3		6.1%			
Majdalany 2021 [36]	Retrospective comparative	URS	Lumenis Pulse™ 120 H	Moses 1.0	230	0.5 J, 50–80 Hz	18	0.94	32 (not compared)	-	10.4	-	-	-	1 month	71	-	17.2% (not compared)			12
				Moses 2.0		0.2–0.3 J, 50–120 Hz	11				14.3					90					
Pietropaolo 2021 [37]	Retrospective Comparative	URS	Lumenis Moses P60 W	Moses	200	0.4–0.8 J, 20–35 Hz	38	10.9	51.6	<0.0001	-	-	-	-	2/4 month	97.3	0.05				16
			Lumenis Holmium 20 W	Regular mode		0.4–0.8 J, 12–18 Hz	38	11.8	82.1		-					81.6					
Wang 2021 [19]	Retrospective Comparative	URS	Lumenis Pulse™ 120 H	Moses Contact	200	0.3 J, 60 Hz	114	12 (990.5)	18.4	0.001	4.99	<0.001	-	-	1 month	86.8	0.743	Fever 3.5%, ARF 4.4%	1.000		15
				Long Pulse			102	12 (993.7)	21.2		5.94					85.3		Fever 4.4%, ARF 3.9%			
Mekayten 2019 [38]	Retrospective Compartive	URS	Lumenis Pulse™ 120 H	Moses	200, 365, 550	0.46 J, 62 Hz (mean)	169	1021	21.1	0.001	3.25	<0.001	-	-	1–1.5 month	87.2	0.469	3.8%	0.225		15
			Dornier Medilas H20 Ho:YAG	Regular mode		0.69 J, 13 Hz (mean)	462	1084	31.8		6.5		-			84.5		6.2%			
**Observation non-comparative studies**																					MINORS scale (0–16)
Reddy 2021 [39]	Prospective	Mini PCNL	Lumenis Pulse™ 120 H	Moses Contact and Moses Distance	365	0.4–0.6 J, 40–60 Hz	110	17.5 (1140)	38.6	-	7.9	-	-	-	1 month	100	-	3.8%	-		12
Leotsakos 2020 [40]	Retrospective	Ultra-mini PCNL	Lumenis Pulse™ 120 H	Moses Contact	550	0.6–0.8 J, 80 Hz	12	31.5 (1252)	93.5	-	12.6	-	-	-	1 month	91.7	-	0%	-		9

Abbreviations: RCT: randomized clinical trial, URS: ureteroscopy, RIRS: retrograde intrarenal surgery, PCNL: percutaneous nephrolithotomy.

## Data Availability

Not applicable.

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
