# Peer review of "New Generation Pulse Modulation in Holmium:YAG Lasers: A Systematic Review of the Literature and Meta-Analysis"

_jcm, 2022, doi:10.3390/jcm11113208_

Round 1

Reviewer 1 Report

Comments to the Author

  • General comments

This is a review article about Holmium YAG laser with pulse modulation which we already used in clinical practice. Pulse modulation like Moses Technology has already been getting familiar in stone management over the world. And then, we got the high-power Holmium laser with Moses and Thulium fiber laser in clinical use now. However, TFL hasn’t expanded yet around the world. This review article focused on the laser with pulse modulation. It is very interesting and needed in now era to summarize those evidence. Therefore, you had better add the meta-analysis in this systematic review to improve your contents of article.

  • Comments for revisions
  • There are some prospective trials. So, how about is the meta-analysis in this systematic review?
  • Thermal injury is one of critical concern in stone management now. There are some published documents regarding with thermal injury. If possible, you should add their evidence in this article.
  • High-power Ho laser with Moses technology has been getting increase use as well as TFL to create much tiny stone dust. If you have some evidence about that, should add and mention about Moses high-power laser with high reputation like 80 Hz.

Author Response

Dear reviewer, thank you very much for your revision. We have included some changes following your report.

1) We added a meta-analysis of the available data 

2) We included comments about thermal injury and the available evidence

3) We inclided comments about High-power Holmium lasers and TFL as competitors to new pulse modulation 

Reviewer 2 Report

This article focused on the lithotripsy performance of new generation PM in Holmium: YAG lasers, holds some clinical significance, but I still have some questions.

1.The authors followed the (PRISMA) recommendations and I think that authors can supplement with a " PRISMA 2020 Checklist " that will be more credible to do so.

2.The authors searched only MEDLINE, Scopus, and Cochrane databases, can more databases be retrieved for more comprehensive results?

3.With respect to the results section, I think that it can't simply list the results of all articles, because the tables already show those results, and more intuitively, I think that this part of the article needs to be substantially modified to make it more clear and concise;

4.The language still needs to be polished.

Author Response

Dear reviewer, thank you very much for your revision. We have included some changes following your report.

1) We added the  PRISMA 2020 Checklist in the supplement

2) We did not search in further databases given that they did not supply more studies to the review

3) We changed the result section and eliminated repeated information

4) We corrected language mistakes

Round 2

Reviewer 2 Report

Dear author, I have read your latest uploaded manuscript. You have used the method of meta-analysis in your latest manuscript. I think the quality of this article has been greatly improved,but maybe it can be more refined in presentation.